



**Bergen Metrics: composite error metrics for assessing performance of climate models**
**using EURO-CORDEX simulations**
Alok K. Samantaray[1,2], Priscilla A. Mooney[1,2], Carla A. Vivacqua[3]
[1]Norwegian Research Centre (Norce), Norway
[2]Bjerknes Centre for Climate Research, Norway
[3]Federal University at Rio Grande do Norte, Brazil
Corresponding author:
Alok Kumar Samantaray
Jahnebakken 5, 5007 Bergen, Norway
Email: asam@norceresearch.no






**Abstract**
Error metrics are useful for evaluating model performance and have been used extensively in
climate change studies. Despite the abundance of error metrics in the literature, most studies
use only one or two metrics. Since each metric evaluates a specific aspect of the relationship
between the reference data and model data, restricting the comparison to just one or two metrics
limits the range of insights derived from the analysis. This study proposes a new framework
and composite error metrics called Bergen Metrics to summarise the overall performance of
climate models and to ease interpretation of results from multiple error metrics. The framework
of Bergen Metrics are based on the p-norm, and the first norm is selected to evaluate the climate
models. The framework includes the application of a non-parametric clustering technique to
multiple error metrics to reduce the number of error metrics with minimum information loss.
An example of Bergen Metrics is provided through its application to the large ensemble of
regional climate simulations available from the EURO-CORDEX initiative. This study
calculates 38 different error metrics to assess the performance of 89 regional climate
simulations of precipitation and temperature over Europe. The non-parametric clustering
technique is applied to these 38 metrics to reduce the number of metrics to be used in Bergen
Metrics for 8 different sub-regions in Europe. These provide useful information about the
performance of the error metrics in different regions. Results show it is possible to observe
contradictory behaviour among error metrics when examining a single model. Therefore, the
study also underscores the significance of employing multiple error metrics depending on the
specific use case to achieve a thorough understanding of the model behaviour.














## 1. Introduction

Climate models are important tools for predicting and understanding climate change, and climate processes (Kotlarski et al., 2014; IPCC, 2021a; IPCC, 2021b; Mooney et al., 2022). In the context of climate studies, climate model evaluation is essential for identifying models that poorly simulate the climate system, and for ranking of climate models (Randall et al., 2007; Flato et al., 2013). The main purpose of climate model evaluation is twofold; firstly, to ensure that the models are reproducing key aspects of the climate system and secondly to understand the limitations of climate projections from the models. This ensures proper interpretation and application of climate models and any climate projections produced by them. The performance of climate models is quantified by different error metrics such as root mean square error, and bias, which assess the agreement between the climate model data and reference data (e.g., gridded observational products, station data, reanalyses, or satellite observations).

Different error metrics are available in the literature, and each has a specific framework according to its purpose (Rupp et al., 2013; Pachepsky et al., 2016; Baker & Taylor, 2016; Collier et al., 2018; Jackson et al., 2019). For example, root mean square error compares the amplitude difference between modelled and reference data, while the correlation coefficient compares the phase difference between modelled and reference data. Depending on the specific error, the error metrics can be categorised into different classes; the most popular classes are accuracy, precision, and association. Accuracy measures the degree of similarity between climate model data and reference data. An extremely high accuracy indicates that the model has less error magnitude of any type and testing the model with other error metrics adds little value (Liemohn et al., 2021). However, if a model has moderate to low accuracy, testing the model with other metrics can reveal other similarities and dissimilarities between model data and reference data. Root mean square error and mean square error are the most used accuracy metrics to evaluate climate models (Watt-Meyer et al., 2021; Wehner et al., 2021; He et al., 2021), even though the metrics cannot reveal whether the model is under or over-predicting the observations. Precision metrics quantify the degree of similarity in the spread of the data. A robust and commonly used metric for assessing the precision of model data is the ratio or difference of standard deviation between modelled data and reference data (van Noije et al., 2021; Wood et al., 2021; Wehner et al., 2021). Finally, association metrics measure the degree of the phase difference between modelled data and observed data. Phase difference is important in climate studies as it affects the initiation and termination time of a season of climate variables. One metric that is extensively used to measure the association is the correlation coefficient (Richter et al., 2022; Bellomo et al., 2021; Yang et al., 2021). Liemohn et al. (2021)



has described various other major categories of metrics and they suggest that assessment of
models should not be restricted to one or two error metrics. Interested readers can follow the
citations to read in detail about the discussed metrics.
There are several composite error metrics that use the modified framework of other metrics to
compute the error magnitude. A widely used example of this is the Taylor diagram (Taylor,
2001), which incorporates correlation, root mean square deviation and ratio of standard
deviation. A distinguishing feature of the Taylor Diagram is its ability to graphically evaluate
the model performance. Another popular example is the Nash-Sutcliffe Efficiency (NSE; Nash
& Sutcliffe, 1970) which is a normalised form of the mean squared error to evaluate and predict
the model streamflow data. Later, it was observed that NSE can be decomposed into three
components which are the functions of correlation, bias and standard deviation (Murphy, 1988;
Weglarczyk, 1998). Other similar scores include the Kling-Gupta (K-G) efficiency (Gupta et
al., 2009) which is a function of three components: ratio of model mean to observed mean, the
ratio of model standard deviation to observed standard deviation and correlation coefficient.
The study of Gupta et al. (2009) argued the NSE, which has a bias component normalised by
the standard deviation of the reference data, will have a low weight on the bias component if
the reference data has high variability. The modified Kling-Gupta efficiency developed by
Kling et al. (2012) involves the ratio of covariance instead of the ratio of standard deviation.
Both K-G efficiency and modified K-G efficiency use Euclidean distance as a basis to calculate
the error magnitude of the model and the study argued that instead of finding a corrected NSE
criterion, the whole problem can be viewed from the multi-objective perspective where the
three error components can be used as separate criteria to be optimised. It identifies the best
models by calculating the Euclidean distance from the ideal point and then finding the model
with the shortest distance. The ideal value of an error metric is obtained when the model exactly
simulates the observed data. The Euclidean distance is also used by Hu et al. (2019) to develop
the DISO metric that incorporates correlation coefficient, absolute error and root mean squared
error. The study of Hu et al. (2019) also argues that accuracy (root mean square error), bias
(absolute error) and association (correlation coefficient) are the three major error classes based
on which a model should be assessed and evaluating a model using a single error metric may
lead to ill-informed results. The study pointed out a few limitations of the Taylor diagram such
as quantification of error magnitude and low sensitivity to small error differences by the
diagram. In a comparative study, Kalmár et al. (2021) found no substantial difference between





DISO index and the Taylor diagram. However, based on quantification of error magnitude,
DISO index can be helpful.
The Euclidean distance framework has been increasingly used in different fields as an error
function or metric for many applications such as evaluation of models, parameter
optimization and classification problems. Euclidean distance is basically the second norm of a
vector. Equation 1 is the generalised form of p-norm in a n-dimensional vector space, where
$x_i$ is the vector. When p is 2, it becomes the Euclidean norm. If the vector ($x_i$) is the difference
between the observed data ($u_i$) and model data ($v_i$) i.e. $x_i = u_i - v_i$, then d is called the
Euclidean distance metric. $i$ represent the time series data. Root mean squared error and mean
squared error are different variants of Euclidian distance metric. If the vector is the difference
between error metrics (correlation coefficient [$u_1$], absolute error [$u_2$] and root mean squared
error [$u_3$]) and their ideal values ($v_{1:3}$), then d is called the DISO index. A disadvantage of the
Euclidean distance is that it suffers the curse of dimensionality (Mirkes et al., 2020; Weber et
al., 1998) i.e. Euclidean distance as a dissimilarity index becomes less efficient as dimension
increases. In this study, we assess the effect of the norm order on the overall error. We use
different measures such as the contribution of outliers to the overall error, the difference
between the maximum and minimum distances, and the average distances to compare different
norms.
$d_n(u,v) = (\sum_{i=1}^{n} |x_i(u_i,v_i)|^p)^{1/p}$         (1)
This study has the following objectives:
i)  Evaluation of 89 CMIP5 driven regional climate simulations from the Euro-
172       CORDEX initiative using 38 error metrics;

ii)  Clustering of error metrics to assess their performance;
iii)  Assessment and recommendation of different p-norms based on their performance;
iv)  Formulation of a composite metric using the optimal norm.
**2.  Data and Study area**
We focus on Europe due to the widespread availability of a large ensemble of high resolution
(0.11°) regional climate simulations. In this study, we use 89 regional climate model (RCM)
simulations from Euro-CORDEX to study the behaviour of different error metrics. The Euro-
CORDEX dataset provides both precipitation and temperature data at 0.11° grid resolution.
The monthly data from 1975 to 2005, which is available in all the RCM simulations, have been
used to calculate the index. Supplementary Table S1 provides an overview of the global climate





models (GCMs) downscaled by the different RCMs. Supplementary Table S2 provides an
overview of the RCMs and assigns a number (Column 1) to each RCM which is used to identify
RCMs in plots that have limited space for labels.
For reference data, both precipitation and temperature data are obtained from E-OBS dataset.
The reference data has a 0.25 º grid spacing. To compare the model data with the reference
data, all the data needs to be on a common grid. In this study, we remapped the RCM data onto
the coarser 0.25 º grid of E-OBS.
The study uses the eight sub-regions of Europe defined by Christensen & Christensen (2007)
– British Isles, Iberian Peninsula, France, Mid-Europe, Scandinavia, Alps, Mediterranean, and
Eastern Europe - to conduct analysis in more homogeneous areas.
**3.   Methodology**
This section outlines the framework for clustering error metrics and provides a brief overview
of their characteristics. Additionally, the section describes the proposed metric's framework.
**3.1 Error metrics**

Error metrics are commonly used in climate change studies to measure the differences between
modelled and reference data in time series. As the number of climate models has increased, the
study of error metrics has become increasingly important. There are several error metrics
available to evaluate the performance of climate models (Jackson et al., 2019), and the selection
of an appropriate metric remains a topic of debate in the literature. For instance, Willmott &
Matsuura (2005) advocate for mean absolute error (MAE) over root mean squared error
(RMSE), as the latter is not an effective indicator of average model performance. In contrast,
Chai & Draxler (2014) contend that RMSE is superior to MAE when errors follow a Gaussian
distribution. To gain insight into the performance of error metrics, we have analysed Euro-
CORDEX precipitation data and examined the differences in ranking of 89 GCM-driven
regional climate simulations using 38 error metrics (Jackson et al., 2019). The list of error
metrics is provided in Table S3. All 89 models are ranked based on their performance using
the 38 error metrics. The average ($r_{M,mean}$; Equation 2) and maximum ($r_{M,max}$; Equation 3)
rank differences are then calculated at each grid point. The former is the mean of all the
pairwise rank differences, while the latter is the maximum of all the pairwise rank differences.
These calculations allow us to understand the performance of different error metrics and the
extent of the disparity in ranking of the climate models.





**Table 1: Example of ranking order**

| Number | Climate model | Ranking order (RO) by $i$th error metric ($E_i$) | Ranking order (RO) by $k$th error metric ($E_k$) |
|--------|---------------|----------------------------------------------------|----------------------------------------------------|
| 1 | M1 | 3 | 2 |
| 2 | M2 | 1 | 3 |
| 3 | M3 | 2 | 1 |


218         $$r_{M,mean} = \mu_g\big(R_{M,k} - R_{M,i}\big) \qquad (2)$$

219         $$r_{M,max} = max_g\,(R_{M,k} - M_{M,i}) \qquad (3)$$

$R_{M,k}$ and $R_{M,i}$ are the rank assigned to model M by the $k$th and $i$th error metric, respectively.
We have provided Table 1 as an example for better understanding of the notations. If there are
three climate models (M1, M2 and M3) as shown in Table 1, all the models have been assigned
to a number (first column) and the order must not change throughout the study. $R_{M,k}$ and $R_{M,i}$
for model M1 are 2 and 3, respectively. $k$ varies from 1 to $N_E$-1 and $i$ varies from $k+1$ to $N_E$,
where $N_E$ is the total number of error metrics. The difference in ranking is calculated for all
possible combinations of error metrics. $\mu_g()$ and $max_g()$ are the mean and maximum operator,
respectively, which is applied across all the grid points (g:1,2,..,gd). gd is the total number of
grid points which is 11370 in this study. Figure 1 demonstrates that different error metrics used
to assess climate models result in significantly different ranking orders. The average of $r_{M,mean}$
across all the grid point varies from 16 to 26 whereas the average of $r_{M,max}$ varies from 40 to
70. The results indicate significant differences in the ranking of the climate models by different
error metrics. The disparity in ranking order may be due to the distinctive error targeted by
each metrics as discussed in the introduction section.
This study assumes that all the errors are important and that it may be necessary to evaluate
model performance using multiple metrics. To achieve independence among the metrics, the
study has attempted to cluster the error metrics based on model performance. This classification
would enable different clusters to have unique characteristics, and metrics within the same
cluster would produce similar results, whereas those from different clusters would yield
different ranking orders. In summary, the study proposes that using multiple error metrics and
clustering them based on performance could improve the understanding and
comprehensiveness of climate model analysis.



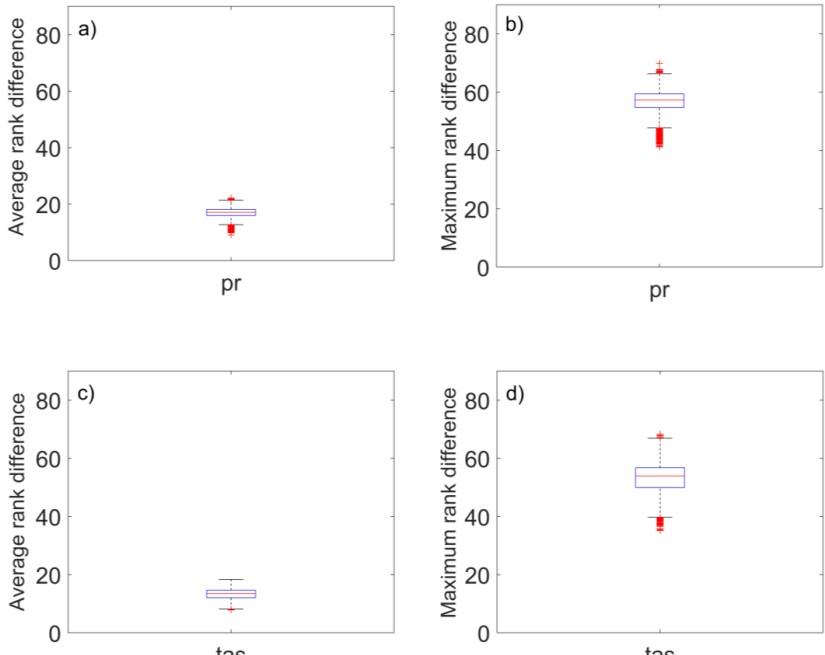

**Figure 1:** Box plot of average rank difference (first column [a, c]) and maximum rank difference (second column; [b, d]) for precipitation (Pr; first row [a, b]) and temperature (T; second row [c, d]) over all the grid points in European region

### 3.2 Clustering of error metrics

The aim of clustering error metrics is to group a set of metrics based on their similarities such that the metrics within the same cluster generate similar rankings of climate models compared to those in different clusters. This study clusters the error metrics using a non-parametric clustering approach inspired by the Chinese restaurant process (CRP; Pitman, 1995). This approach was chosen based on its performance compared to the k-means clustering approach (see Text S1) and its simpler framework. The algorithm follows two fundamental principles: (i) the first error metric ($E_1$) forms the first cluster ($C_1$), and (ii) the ith error metric ($E_i$) is assigned to a cluster which has the maximum of all the mean absolute error ($u_j$) values greater than a particular threshold value (th). The clustering algorithm is presented in Fig. 2.

Similar to the rank difference explained in the previous section, the MAE ($RO_i, RO_k$) between the ranking order produced by two error metrics is computed. RO is the ranking order and it can be calculated by assigning the climate models to a number. For example, the ranking order ($RO_i$) by ith error metric and the ranking order ($RO_k$) by kth error metric are [3, 1, 2] and [2, 3, 1], respectively in Table 1. The MAE values are calculated for all possible combinations of



error metrics in a particular cluster and the maximum of the MAE values is used to compare it
to the threshold value. The exercise is repeated for all the clusters ($N_C$) available at that time.
The number of clusters ($N_C$) and the number of error metrics in each cluster ($N_{CE}$) are updated
for each iteration (i) and if the criteria is not satisfied, then a new cluster is formed using that
error metric. The whole exercise is repeated till all the error metrics ($N_E$) gets assigned to a
cluster.

$E_1 \in C_1$                                 First error metric belongs to the first cluster

For i = 2:$N_E$ do                       For all the error metrics

   For j < $N_C$ do                       For all the clusters

      For k < $N_{CE}$ do      For all the error metrics in $C_j$

      $U_{j,k} = MAE(RO_i, RO_k)$

   $u_j = max(U_{j,k})$

   If   $u_j$ < th

   $E_i \in C_j$

   else

   $E_i \in C_{N_c+1}$


**Figure 2:** Algorithm of the non-parametric clustering for classifying the error metrics
The threshold value is defined as qth percentile of a column matrix D where D is the collection
of MAE values for all possible combinations of error metrics at all the grid points in a region.
In this study, q has been assigned the value of 10 and the sensitivity of q is discussed in the
results section.
**3.3 Proposed metric- The Bergen Metrics**
The clustering of error metrics guarantees that metrics in different groups produce distinct
ranking orders, implying that each group targets different errors. One of the objectives of this
study is to integrate different errors and create a composite error to obtain a single value. One
potential solution is to use the Euclidean distance approach with different error metrics as
different dimensions in the Euclidean space. To illustrate this, we employed three widely used
error metrics: Normalized Root Mean Square Error (RMSE), Standard Deviation ratio (SD)
and correlation coefficient. In the Euclidean space, an ideal model that predicts the climate
variable as accurately as the observed data would have values of 1, 1, and 0 for correlation
coefficient, Standard Deviation ratio, and normalized RMSE, respectively. The coordinates of



an ideal model in the Euclidean space would be (1, 1, 0), as represented by the red point in Fig.
3a. Since different models have unique coordinates based on the three metrics, these
coordinates serve as possible solutions to determine the best model. If a decision is required,
one approach could be to calculate the Euclidean distance from the ideal point to all points and
select the point with the shortest distance (Equation 4). This equation can be simplified to
Equation 5. The model that is closest to the ideal point, indicated by the optimal point in Fig.3b,
can be considered as the best model.
$$ED\ Metric\ =\ \sqrt{\begin{aligned}(1-Correlation\ coefficient)^2 + (1-Standard\ deviation\ ratio)^2\\+(0-RMSE)^2\end{aligned}}\quad(4)$$
$$ED\ Metric\ =\ \sqrt{\begin{aligned}(1-Correlation\ coefficient)^2 + (1-Standard\ deviation\ ratio)^2\\+(RMSE)^2\end{aligned}}\quad(5)$$

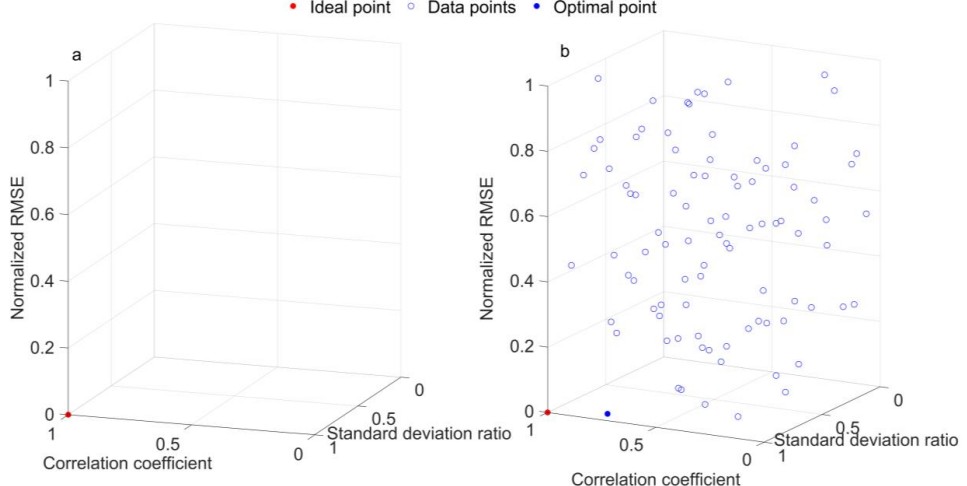


**Figure 3:** Example for three-dimensional (a) ideal point and (b) the solution space      of
correlation coefficient (x-axis), standard deviation (y-axis) and normalized RMSE (z-axis)

The Euclidian distance has several benefits that make it a popular metric, primarily its
simplistic framework. However, it also has some drawbacks. The Euclidian distance, also
known as L2 norm, is less effective in higher dimensional spaces, which can lead to instability
when additional error metrics are added (Weber et al., 1998; Aggarwal et al., 2001). To mitigate
this issue, recent research has focused on the use of L1 norms, such as relative mean absolute
error and mean absolute scaled error, which have become more popular than L2 norms like
mean squared error. This approach reduces the impact of outliers in the data (Armstrong &



Collopy, 1992; Hyndman and Koehler, 2006). Reich et al. (2016) found that relative MAE,
based on an L1 norm, is advantageous in assessing prediction models. This study proposes the
following new metrics called the Bergen Metrics (BM) which is a generalised p-norm
framework to evaluate climate models. Equation 6 presents the generalised form of the metric.
It is important to note that equation 6 serves as an illustration of Bergen metrics, and users
have the flexibility to include or remove metrics according to their preference.
$$Bergen\ Metric\ =\ \sqrt[p]{\begin{array}{c}(1 - Correlation\ coefficient)^p\\ +(1 - Standard\ deviation\ ratio)^p\\ +(0 - RMSE)^p\end{array}}\quad (6)$$

A case study has been conducted to understand the impact of different p norms on the ranking
order of climate models. For this, five error metrics - RMSE, bias, correlation coefficient,
standard deviation ratio, and mean ratio - have been considered (Equation 7) and the error
metrics are normalised using model data. The study includes 89 RCM simulations for
precipitation, and Fig. 4a shows the ranking of these models for different p norms. The lines
corresponding to each model give information about the model's ranking in different norms.
The results demonstrate that climate models are highly sensitive to p norms. Significant change
in ranking order is observed for the first four norms. Fig. 5 shows the percentage contribution
of outliers to the total error magnitude for models that have outliers. Median absolute deviation
technique (MAD) is used to identify outliers among the error metrics. Some of the models
have only one outlier (plots with a single solid line in Fig. 5) and other models have two outliers
(plots with both solid and dotted lines in Fig. 5). The percentage contribution of outliers
increases as the p norm increases, consistent with previous literature (Armstrong and Collopy,
1992; Hyndman and Koehler, 2006). The study has used two parameters to indicate the
capability of each norm to differentiate between climate models - mean pairwise difference of
the BM and the difference between the maximum and minimum values of the BM. Figure 4b
shows that both parameters decrease as the p norm increases, indicating less differentiability.
The results suggest that the first norm (p=1) is the optimal norm to use as a metric in this study
and will be utilized in the following analyses.
$$Bergen\ Metric\ (BM)\ =\ \sqrt[p]{\begin{array}{c}(0 - RMSE)^p + (0 - Bias)^p\\ +\ (1 - Standard\ deviation)^p\\ +(1 - Correlation\ coefficient)^p + (1 - Mean\ ratio)^p\end{array}}\quad (7)$$






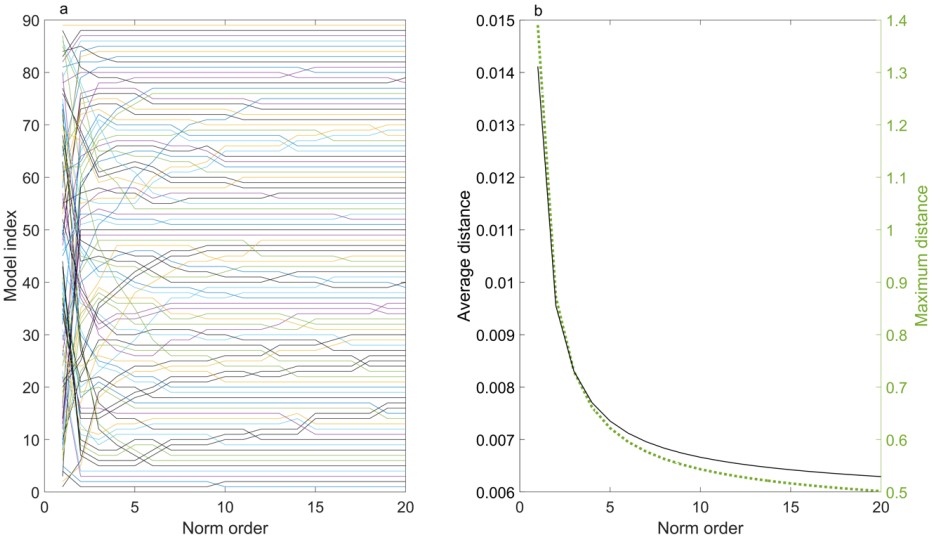

**Figure 4:** a) The change in the ranking of the climate models with different norm order (p) b) the change in the difference between the maximum and minimum distances and the average distances with different norm order

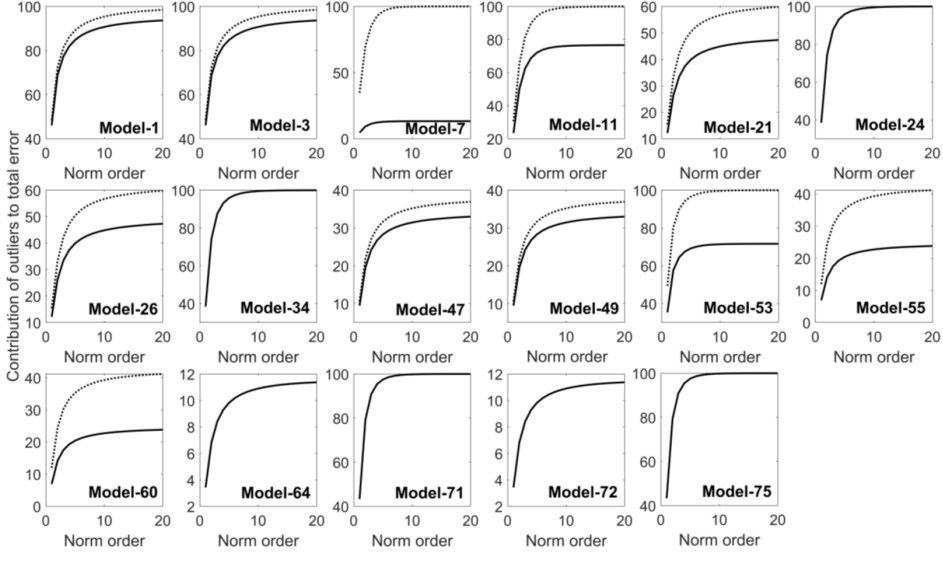

**Figure 5:** The percentage contribution of outliers to the total error magnitude as a function of norm order. The colours represent different outliers.





## 4. Results

### 4.1 Regional clustering of error metrics

The study considers 38 error metrics (Table S3) which can take both positive and negative values as input. Similar to the models, the error metrics have been assigned a number (column 1; Table S3) and the error metrics have been labelled as those numbers in some figures.

The clustering technique described in the methodology section can be applied to individual grid points, but for the sake of simplicity, we use a single cluster for all grid points within each of these regions defined by Christensen & Christensen (2007). The methodology is modified slightly to enable regional clustering. At a grid point scale, the maximum value of mean absolute error ($u_j$) is used as a proxy for that specific error metric at a grid point. For regional clustering, the maximum MAE values are computed for all grid points within the region, and the average of those values is used as a proxy for that region and error metric. This value is then compared with a threshold to determine whether the error metric belongs to a certain cluster or it should be assigned to a new cluster. The clustering algorithm is executed for multiple thresholds.

The 5th, 10th, and 20th percentiles are selected as potential thresholds to cluster the error metrics. However, users can select any number of thresholds for the sensitivity analysis. The clustering algorithm is allowed to run for all the thresholds to determine the optimal threshold. The efficiency of each cluster for a given threshold is represented by the mean of MAE over all the clusters. Another criterion used to determine the threshold is the number of clusters corresponding to each threshold. An increase in the percentile (q) is expected to increase the MAE as the magnitude of threshold increases. Similarly, the number of clusters are expected to decrease as q increases as it can allow more error metrics into a cluster due to higher threshold magnitude. From Fig. 6, we conclude that the results are according to our expectations. It is found that increasing the percentile resulted in an increase in MAE and a decrease in the number of clusters. The 10th percentile is selected as the threshold to cluster the error metrics for both temperature and precipitation, as it has a smaller number of clusters compared to 5th percentile and less MAE compared to 20th percentile. The



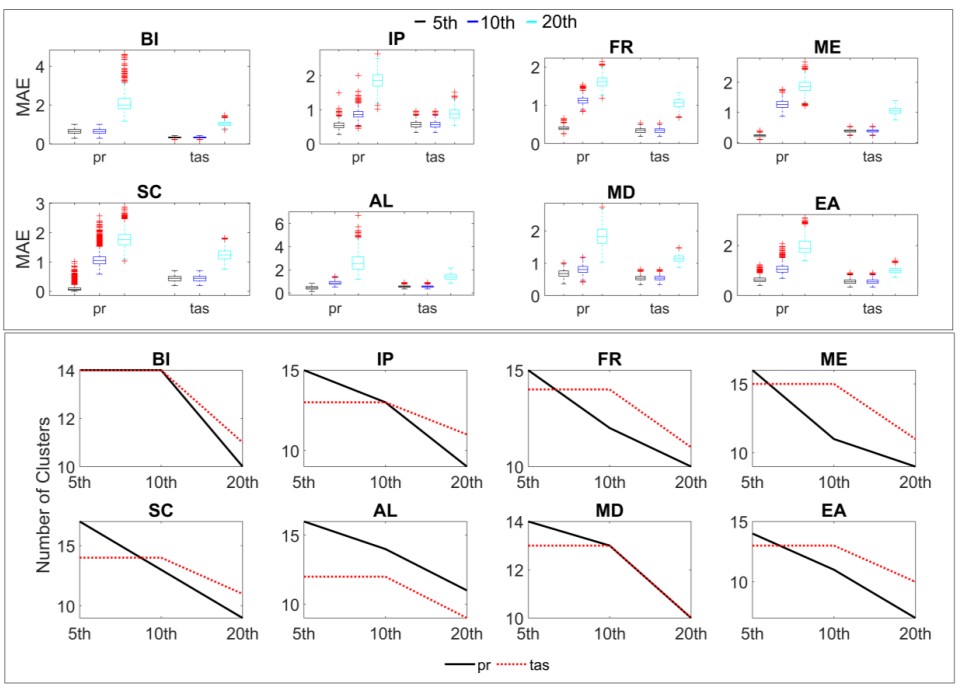

**Figure 6:** The variation in MAE (first box) and number of clusters (second box) corresponding to 5th, 10th and 20th percentile for precipitation (pr) and temperature (tas) for all the eight regions

## 4.2 Results of clustering

### 4.2.1 Precipitation

For the British Isles region, the classification of 38 error metrics resulted in 15 clusters, with 8 error metrics being single point clusters due to their unique behaviour (Fig. 7). These 8 metrics are d [2], (MB) R [17], MdE [19], MEE [21], MV [22], r2 [31], SGA [35], and R(Spearman) [36]. The threshold for precipitation data is 6.35, indicating that all 8 error metrics produced MAE values greater than 6.35 compared to the remaining 30 error metrics. RMSE [32] and its variants such as normalized RMSE by IQR [25], mean [26] and range [27] are assigned to the same cluster, as ED [7], IRMSE [9], MAE [13], MAPD [15], MASE [16], and MSE [23]. The reason could be the L-norm framework which is used by most of the error metrics in this cluster. D1 [3], d1 [4], and d(Mod.) [5] which share a similar framework, are also assigned to a single cluster. Error metrics that evaluate the phase difference between observed and modelled data, including ACC [1], R (Pearson) [30], SC [34], and M [38], are assigned to a single cluster. H10(MAHE) [8] and MALE [14] share the same cluster as both metrics consider the difference of logarithmic of the model and observed data to compute the error. Similarly, MdAE [18] and



MdSE [20] are assigned to a single cluster, as both metrics use the median of the difference
between observed and modelled data. However, MdE [19] is assigned to a different cluster as
it only considers the difference between observed and modelled data without bringing them to
the positive domain. NED [24] and SA [33] are found to be in the same cluster, as both metrics
are linearly associated while evaluating the model, even though their underlying frameworks
are somewhat different. Although ED [7] and NED [24] follow the L2 norm, they are not
assigned to the same cluster. This can be attributed to the normalisation of observed and
modelled data by their respective means in NED, as the statistical parameters such as mean is
sensitive to outliers, which can result in changes in ranking order.

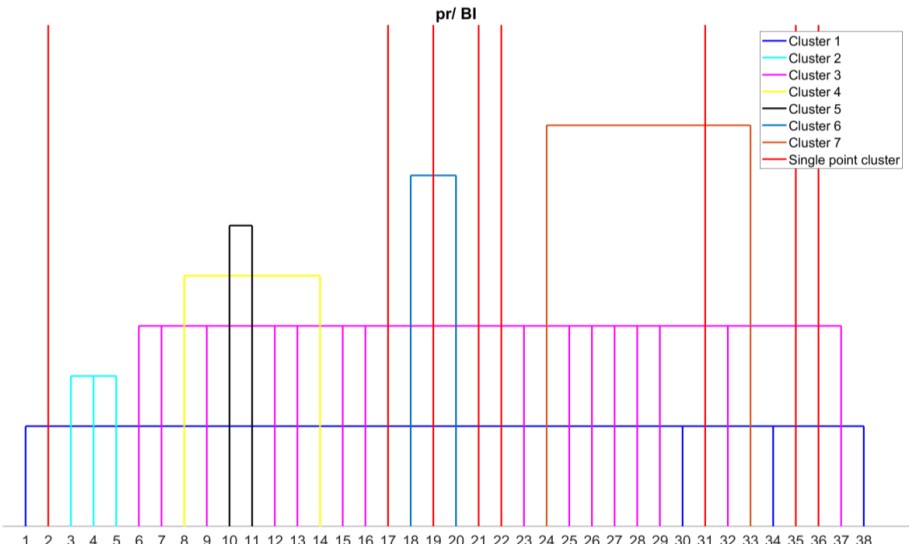


**Figure 7:** Clustering of error metrics using precipitation (pr) data for British Isles (BI) region.
Each error metric can be identified by the number using Table S3.
The Iberian Peninsula region is found to have 17 clusters, with 12 of them being single point
clusters (Fig. 8). Seven of the eight error metrics that are single point clusters in British Isles
are also single point clusters in Iberian Peninsula, except for r2 [31]. Five other error metrics:
NED [24], KGE (2009) [10], KGE (2012) [11], SA [33], and M [38] are also single point
clusters in Iberian Peninsula region. In British Isles,  KGE (2009) [10] and KGE (2012) [11]
are assigned to the same cluster. The KGE (2012) is different from KGE (2009) since it used
the ratio of coefficient of variation between modelled and observed data instead of the ratio of
standard deviation to avoid the cross-correlation between bias and variability ratio. The



coefficient of variation is the ratio between the standard deviation and the mean of the data,
which represents the extent of variability with respect to the mean of the data. A biased dataset
can produce a significant change in the relative standard deviation, i.e., the coefficient of
variation. That is a possible reason why both the metrics are in different clusters. r2 is assigned
to the correlation metrics cluster in this region. The remaining clusters are almost identical to
the clusters obtained for the British Isles region.

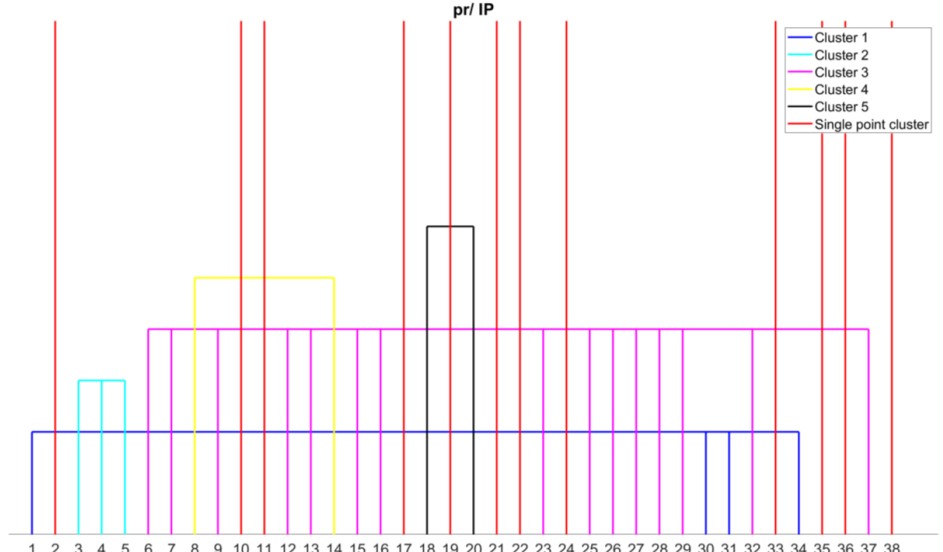


**Figure 8:** Clustering of error metrics using precipitation (pr) data for Iberian Peninsula (IP)
region. Each error metric can be identified by the number using Table S3.
As the results for the other 6 regions are similar to either the British Isles or the Iberian
Peninsula, we simply summarise their results here and refer the reader to the supplementary
material for further information. France (Fig. S2), Mid-Europe (Fig. S3), Scandinavia (Fig.
S4), Alps (Fig. S5), Mediterranean (Fig. S6) and Eastern Europe (Fig. S7) exhibit 15, 15, 16,
16, 17, and 14 clusters, respectively, with 8, 8, 10, 10, 12, and 6 single point clusters. France
and Mid-Europe have the same clusters as the British Isles, and the Mediterranean has the same
clusters as Iberian Peninsula. Scandinavia has clusters similar to British Isles, except that M
[38] is a single point cluster and r2 [31] has been assigned to the correlation metrics cluster in
Scandinavia.  The Alps also has clusters similar to British Isles, except  KGE (2009) [10] and
KGE (2012) [11] are single point clusters. Eastern Europe also has clusters similar to British



Isles, with the exception that d [2], which is a single point cluster in British Isles, forms a new
cluster with M [38] in Eastern Europe.
**4.2.2 Temperature**
Compared to precipitation data, temperature data has a lower number of clusters, which can be
attributed to the lower variability in temperature data. The clustering of error metrics for British
Isles is shown in Fig. 9. For British Isles, 12 clusters are identified, with 5 single point clusters,
namely KGE(2009) [10], KGE(2012) [11], MV [22], SGA [35], and R(Spearman) [36]. Similar
to precipitation clusters, several error metrics, including ED [7], IRMSE [9], MAE [13], MAPD
[15], MASE [16], MSE [23], NRMSE(IQR) [25], NRMSE(mean) [26], NRMSE(range) [27]
and RMSE [32] are assigned to the same cluster.

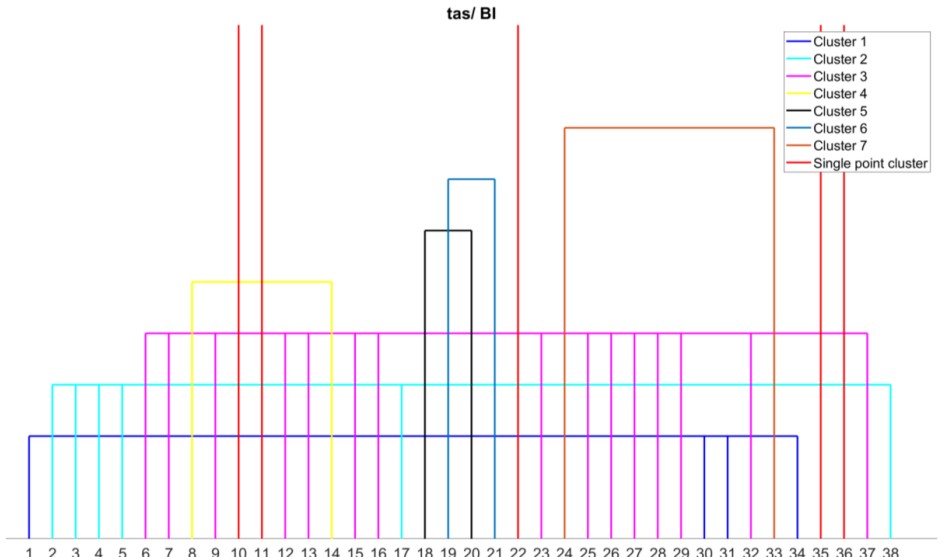


**Figure 9:** Clustering of error metrics using temperature (tas) data for British Isles (BI) region.
Each error metric can be identified by the number using Table S3.
The correlation metrics, such as ACC [1], r2 [31], SCO [34], and R(Pearson) [36] belong to
the same cluster. France (Fig. S8) and Mid-Europe (Fig. S9) have the same cluster as British
Isles for temperature data. For Iberian Peninsula (Fig.10), 13 different clusters are identified,
with 7 single point clusters, including MdE [19] and MEE [21] in addition to the 5 single point
clusters from British Isles. The remaining clusters are similar to those in British Isles.
Mediterranean (Fig. S10) has the same cluster as Iberian Peninsula for temperature data, with
13 clusters and 7 single point clusters. Scandinavia (Fig. S11) and Eastern Europe (Fig. S12)



have the same number of clusters i.e. 14 clusters. Scandinavia has 8 single point clusters
whereas Eastern Europe has 9 single point clusters. Alps (Fig. S13) has 15 clusters with 10
single point clusters.

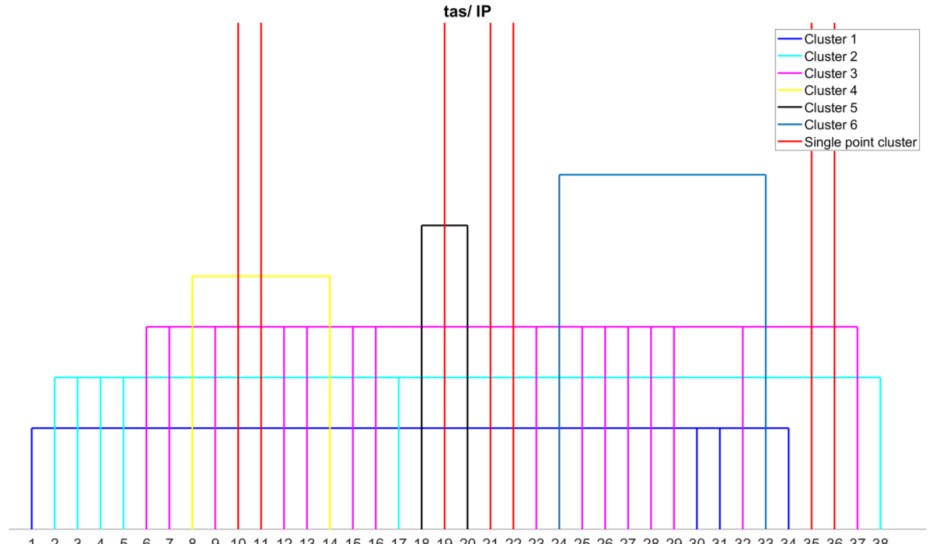


**Figure 10:** Clustering of error metrics using temperature (tas) data for Iberian Peninsula (IP)

region. Each error metric can be identified by the number using Table S3.

### 4.3 Bergen Metrics

A Bergen metric is computed for all eight regions using the respective clusters for both
precipitation and temperature. A single metric is chosen from each cluster randomly; Random
selection demonstrated no discernible impact on the ranking (see Supplementary Material).
Although computed for all 89 regional climate models, this paper focuses on discussing only
one climate model for both precipitation and temperature. The CLM Community (CLMCom)
regional model from ICHEC-EC-EARTH for r3i1p1 realisation is discussed as it performed
best at over 25 grid points in 5 regions and more than 2 grid points in seven regions. For the
temperature variable, the CLMCom model form CCCma-CanESM2 model for r1i1p1
realisation is discussed, as it performed best at over 25 grid points in seven regions.

### 4.3.1 Precipitation

A Bergen metric (BM) is used to assess the performance of the CLMCom model for
precipitation in all eight different regions. The BM in British Isles region is a composite metric
that takes into account 15 different error metrics i.e. ACC, D1, dr, H10(MAHE), KGE(2009),



MdAE, NED, d, MB(R), MdE, MEE, MV, r2, SGA, and R(Spearman). Figure 11 provides an

overview of the spatial distribution of the BM for all eight regions, while the spatial distribution

of each of these metrics is shown in Fig. 12 for the British Isles region.

The magnitude of BM ranges from 0 to 13, with a score of 0 indicating good performance by

the model. Based on the results, the CLMCom model performed well in the western part of

British Isles, as indicated by the BM. This is a result of the good performance of most of the

individual metrics that comprise the Bergen Metric. This is shown in Fig. 12. There are some

contradictory results from different error metrics in the eastern region. While all 13 metrics

indicate good performance, the MV, r2 and NED indicate very bad performance by the model.

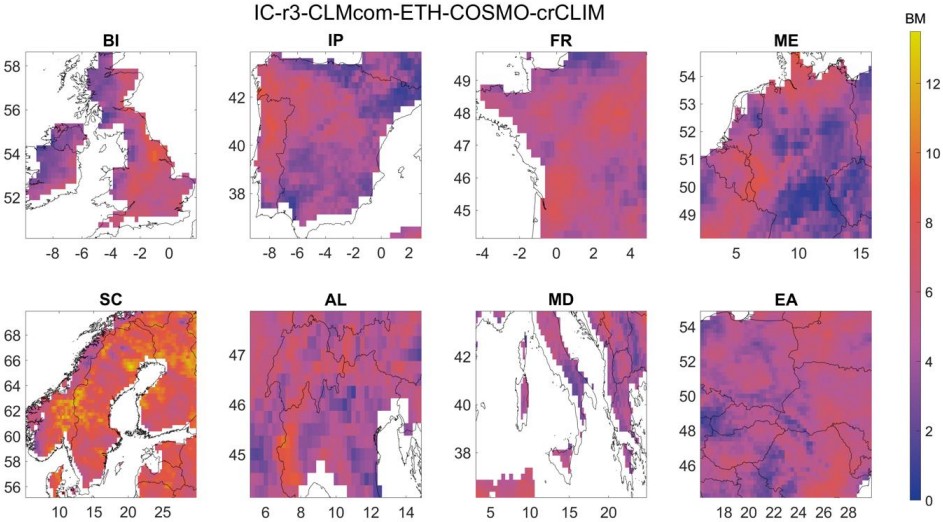

**Figure 11:** Spatial distribution of Bergen metric using precipitation data for all the eight

regions



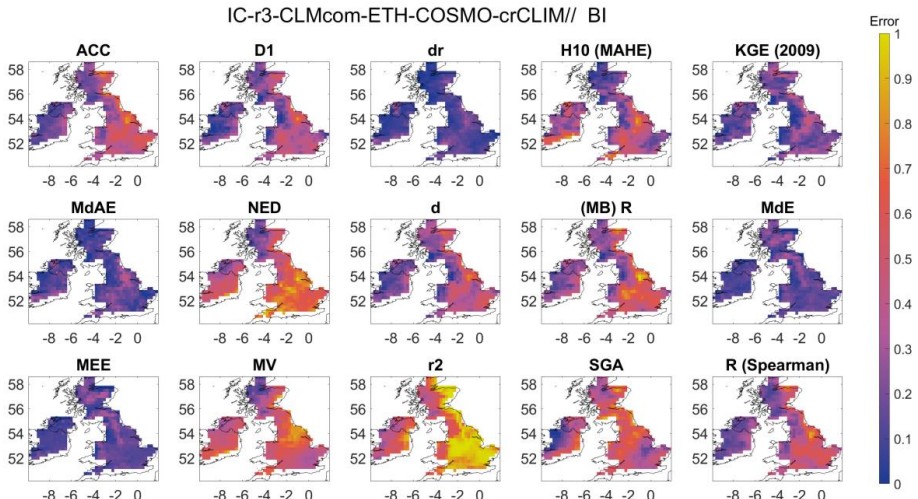

**Figure 12:** Spatial distribution of the error metrics used to compute the Bergen metric for precipitation and for British Isles (BI) region. The error metrics have been labelled by the abbreviation and the corresponding error metrics can be identified from Table S3.

The use of individual error metrics can provide meaningful insights into the performance of the model in different regions. For example, metrics such as dr, MdAE, MdE, and MEE indicate good performance in the southeastern region, while R(Spearman) indicates bad performance by the CLMCom model which implies that the phase difference is significant between observed and modelled data in this region. It is worth noting that some metrics, such as r2 and R(Spearman), may provide different results even though they share a similar framework. R(Spearman) only tells how well the modelled data follow the observed data while r2 indicate how well the data represents the line of best fit (https://tinyurl.com/y52r3xed; https://tinyurl.com/yk2jmsxt). Overall, the use of multiple error metrics and the analysis of individual metrics can provide a more comprehensive assessment of the model's performance, particularly in regions where different metrics provide conflicting results.



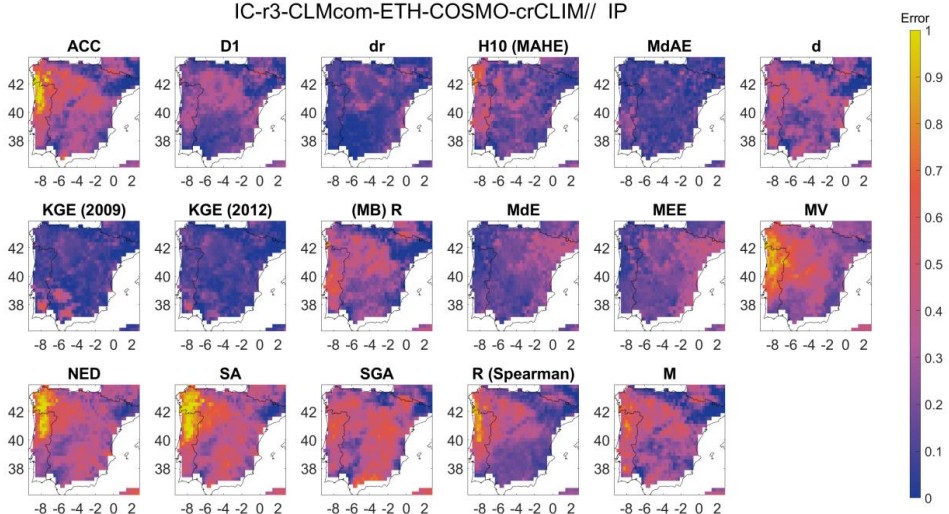

**Figure 13:** Spatial distribution of the error metrics used to compute the Bergen metric for precipitation and for Iberian Peninsula (IP) region. The error metrics have been labelled by the abbreviation and the corresponding error metrics can be identified from Table S3.

Figure 14 shows a Bergen metric for Iberian Peninsula applied to the CLMCom model, which is based on 17 error metrics obtained from each cluster. These metrics, including ACC, D1, dr, H10 (MAHE), MdAE, d, KGE (2009), KGE (2012), MB (R), MdE, MEE, MV, NED, SA, SGA, R (Spearman) and M, are presented in Fig. 13. The results indicate that the model performs relatively better in the northeast and southeast regions compared to the western region (see Fig. 11), possibly due to the influence of certain metrics such as ACC, R (Spearman), MV, NED, and SA. Additionally, while KGE (2009) and KGE (2012) exhibit similar spatial error patterns, further analysis in the southern region reveals the differences in the magnitude of error. Interestingly, despite their similarity, KGE (2009) and KGE (2012) are classified into different clusters based on a threshold MAE of 5.41, used to determine cluster membership.

France (Fig. S14), and Mid-Europe (Fig. S15) have the same clusters as the British Isles, and therefore the same error metrics used in British Isles are used to calculate the Bergen metric for France and Mid-Europe. The Bergen metric indicates an average performance of the model for the entire study region of France (see Fig. 11). While r2 shows a very poor performance of the model for France, MEE metric shows a completely opposite trend, indicating a very good performance of the model. Similar disagreement between r2 and MEE is also observed in the



British Isles. On the other hand, SGA, which compares the shape of the two signals, shows an
average performance by the model. In terms of the spatial distribution of error, the Bergen
metric shows lower error magnitudes for MEE in the southeast part of the study region.
The Bergen metric is also used to assess the performance of the CLMCom model for
Scandinavia and Alps using 16 error metrics from each cluster, including ACC, D1, dr, H10
(MAHE), MdAE, NED, d, KGE (2009), KGE (2012), MB (R), MdE, MEE, MV, SGA, R
(Spearman) and M. The spatial distribution of these metrics is presented in supplementary Fig.
S16 (Scandinavia) and Fig. S17 (Alps).
Fig. S16 and Fig. 11 suggest that the CLMCom model does not perform well for Scandinavia.
However, some error metrics, including dr, MdAE, MdE, and MEE, show good performance
in the southern part of the region. Although MdAE, MdE, and MEE are assigned to different
clusters, they exhibit similar spatial distributions of error. It is worth noting that despite the
similarity, the three error metrics are in different clusters due to their higher MAE between
them. For the Alps, the Bergen metric indicates a relatively good performance of the CLMCom
model. It can be observed in Fig. S17, all metrics except r2 show good performance for the
model.
The Mediterranean has the same clusters as the Iberian Peninsula, and the spatial distribution
of each metric for the Mediterranean is presented in Fig. S18. The Bergen metric for the
CLMCom model suggests an average performance for the entire Mediterranean region. Some
of the error metrics, such as KGE (2009), KGE (2012), dr, and MdAE, indicate good model
performance. However, metrics such as SGA, SA, and NED, show relatively poor performance
of the model.
For  Eastern Europe, the Bergen metric is computed using 14 error metrics from each cluster,
as listed: ACC, d, D1, dr, H10(MAHE), KGE(2009), MdAE, NED, MB(R), MdE, MEE, MV,
SGA, and R(Spearman). The spatial distribution of each metric is presented in Fig. S19. One
notable observation from the figure is the difference between SGA and MEE, which indicates
that although the model data has a low bias, the direction of error of the modelled data is
completely different from that of the observed data. This insight can be valuable in identifying
areas where the model's performance can be improved.
**4.3.2 Temperature**
For temperature, we focus on the CLM Community (CLMCom) regional model driven by
ICHEC-EC-EARTH to demonstrate the application of Bergen metrics for temperature. The
spatial distribution of BM is shown in Fig. 14, which indicates average performance by the



model, except in certain areas like northern part of Scandinavia, central part of Eastern Europe
and western part of Iberian Peninsula, where the performance is bad. The British Isles (Fig.
15), France  (Fig. S20), and Mid-Europe (Fig. S21) regions have 12 clusters, and 12 error
metrics, including ACC, d, dr, H10(MAHE), MdAE, MdE, NED, KGE(2009), KGE(2012),
MV, SGA, and R(Spearman) are used to compute the Bergen metric for these regions.

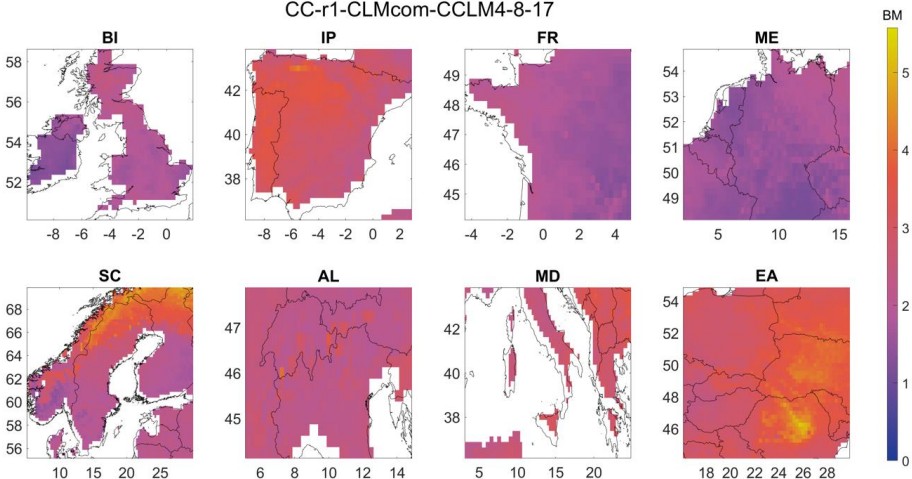


**Figure 14:** Spatial distribution of Bergen metric using temperature data for all the eight regions
The Scandinavia (Fig. S22) and Eastern Europe (Fig. S23) regions have 14 clusters and all the
error metrics from British Isles, along with VE and SA, are used to compute the Bergen metric
for these regions. The Iberian Peninsula (Fig. 16) and Mediterranean (Fig. S24) regions have
the same cluster, with a total of 13 clusters and all the error metrics from British Isles, plus
MEE, are used to compute the Bergen metric. The Alps (Fig. S25) region has 15 clusters, with
all the error metrics from Scandinavia, including MEE, used to compute the Bergen metric.
MdE and MEE consistently indicate very bad model performance for all the regions, while the
other metrics indicate relatively good performance. This suggests that the mean and median of
the modelled data tend to underestimate/overestimate the observed mean and median,
respectively. Histograms in Fig. 17 further investigate this, showing that the error values for
ACC are more evenly distributed in the Iberian Peninsula region and close to its ideal point 1,
while the source errors for MdE and MEE are concentrated between -0.5 to -1.5, resulting in
most of the error values being concentrated between 0.9 to 1 after normalization. The source
error represents the distance between the ideal values and actual magnitude after normalization.
Similar patterns can be observed in the other regions for temperature    .



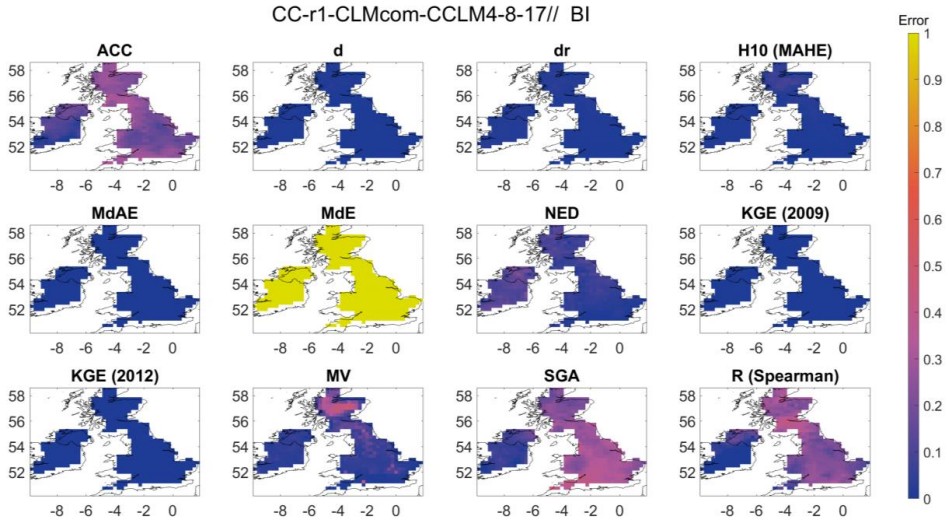

**Figure 15:** Spatial distribution of the error metrics used to compute the Bergen metric for temperature and for British Isles (BI) region. The error metrics have been labelled by the abbreviation and the corresponding error metrics can be identified from Table S3.

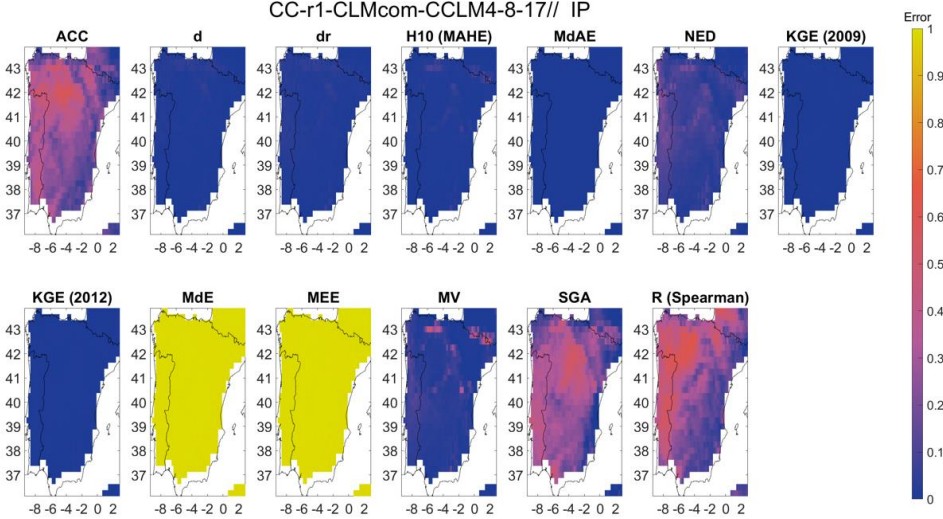

**Figure 16:** Spatial distribution of the error metrics used to compute the Bergen metric for temperature and for Iberian Peninsula (IP) region. The error metrics have been labelled by the abbreviation and the corresponding error metrics can be identified from Table S3.



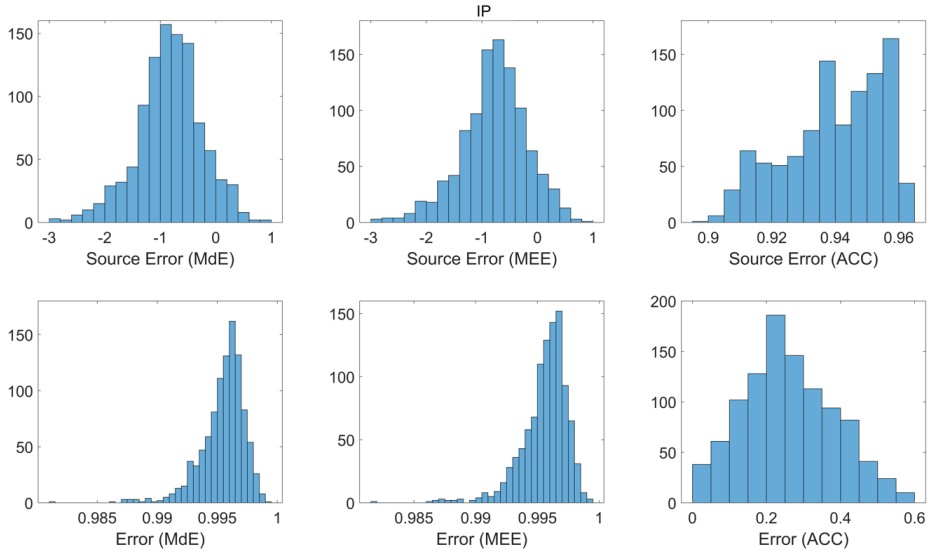

**Figure 17:** Histogram plot of error and source error for MdE, MEE and ACC for Iberian Peninsula region (IP).

## 5. Conclusions

A framework of new error metrics, known as 'Bergen metrics', has been introduced in this study to evaluate the ability of climate models to simulate the observed climate through comparison with a reference field. The proposed metric integrates several error metrics, as described in the results section. To generate a single composite index, the methodology uses a generalized p-norm framework to merge all the error metrics. The research determines that the first norm is the most effective norm to use in the analysis.

The study also shows that the number of error metrics used in Bergen Metrics can be reduced using a non-parametric clustering technique. Although several clustering techniques are already available in the literature, they come with certain requirements. Either they require the number of clusters before running the algorithm or information on the class label of the feature vector. The adopted clustering technique tries to identify the natural cluster present in the data. The mean absolute error based on ranking order is used as a dissimilarity index to assign error metrics to different clusters. The technique also has a threshold parameter $5^{th}$, $10^{th}$ and $20^{th}$ are selected as candidates for threshold parameter and $10^{th}$ percentile of the D matrix is adopted as a threshold in this study. It is selected because increase in threshold ($20^{th}$ percentile) resulted in increase in MAE and decrease in number of clusters, whereas, decrease in threshold ($5^{th}$





percentile) resulted in decrease in MAE and increase in number of clusters and the study chose
a middle ground. However, users can investigate different values of q before choosing the
threshold. The clustering technique is compared with the K-means clustering approach and it
is found that the non-parametric technique has lower MAE compared to the K-means approach.
The clustering is performed for all the eight regions and those are British Isles, Iberian
Peninsula, France, Mid-Europe, Scandinavia, Alps, Mediterranean and Eastern Europe. For
precipitation, 15, 17, 15, 15, 16, 15, 17, and 14 clusters are obtained for the eight regions,
respectively. For temperature, 12, 13, 12, 12, 14, 15, 13, and 14 clusters are obtained for the
eight regions, respectively.
A single error metric from each cluster can be chosen randomly as a component to be used in
the calculation of a Bergen Metric. We have shown that random selection does not have any
effect on the ranking order produced by a Bergen Metric. The Bergen Metric which uses the
L1 framework is found to be less sensitive to outliers compared to the other norms and more
stable in higher dimensional space. Bergen Metrics are a multivariate error functions that can
take any number of error metrics of different variables as shown in the last section. It can be
further modified for a weighting-based metric that can allow the user to give more weightage
to particular metrics depending on the requirement of the study. While some metrics show good
performance in certain regions, others indicate poor performance. It is also important to observe
how a single metric can influence and change the ranking of climate models. Bergen metrics
provide a comprehensive evaluation of the model's performance, which is useful for identifying
the strengths and weaknesses of the model in different contexts.
Future research should address the sampling uncertainty associated with Bergen metrics. Each
data point in time series data has a certain contribution to the total error and if the contribution
is not evenly distributed for all the data points, the metric may give biased results. Also, each
metric has probabilistic uncertainty associated with it. For example, RMSE works well when
the errors are normally distributed and what if the errors are not normally distributed.
Discussion on uncertainty may yield useful information that will be helpful in removing the
bias from climate models in the future.





**Data and Code availability**

The EURO-CORDEX data used in this work are obtained from the Earth System Grid

Federation server. The reference precipitation and temperature data is available at

*https://cds.climate.copernicus.eu/cdsapp#!/dataset/reanalysis-era5-pressure-levels-monthly-*

*means-preliminary-back-extension?tab=form*

The code for clustering the error metrics is available at   https://github.com/badal01/Error-

metrics-clustering.

**Author contributions**

AS developed the methodology and performed the formal analysis. PM supervised the research

activity planning and execution. AS prepared the first draft of manuscript. All authors

contributed to editing and reviewing the manuscript.

**Competing interests**

The authors declare that they have no conflict of interest.

**Acknowledgements**

 The FRONTIER project has received funding from the Research Council of Norway (project

number 301777). We thank James Done and Andreas Prein for their advice and critical

comments regarding the work.















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
