# Peer review of "Bergen Metrics: composite error metrics for assessing performance of climate models using EURO-CORDEX simulations Alok K. Samantaray1,2, Priscilla A. Mooney1,2, Carla A. Vivacqua3"

_Geoscientific Model Development, 2023_

## Author Comment (AC3)

**Bergen Metrics: composite error metrics for assessing performance of climate models using EURO-CORDEX simulations**
Alok K. Samantaray, Priscilla A. Mooney, Carla A. Vivacqua

We appreciate the comments from the reviewers as they have made some important points that have been addressed in the revised manuscript.

I enjoyed reading the ms. However, I have several concerns listed below:

We thank the reviewer for the comment. In the revised manuscript, we have addressed all the comments suggested by the reviewer.

L65: "38 different metrics" the reader can be curious where are they or how they are obtained from Eq7? Explain in more detail on section 3.3.

We thank the reviewer for the comment. In this study, we employed a clustering approach to group the 38 different error metrics. Randomly selecting one metric from each cluster, we utilized them in Equation 7 to calculate the Bergen metric. For enhanced clarity on this process, we have incorporated a flowchart in the revised draft. This flowchart provides a step-by-step illustration of how the Bergen metric is computed using different error metrics.

[Figure]

Figure R1: The flowchart for the calculation of Bergen metric

Eq 4-5-6-7: listing all four forms of the BM is confusing only Eq7 is enough for the reader.

We appreciate the reviewer's comment. As the number of metrics employed in Equation 4, 5 and 6 differs from Equation 7, we have retained Equation 4, and 7 while removing Equation 5 and 6.

Eq7: none of the components are innovative or directly related to distribution. They are all bias-sensitive components and may have overlapping (redundant) information.

We appreciate the reviewer's feedback. In the draft, we explicitly state that our study utilizes established error metrics, examining their focus on specific aspects of model and reference data and how they contribute to the ranking of climate models. A crucial assumption in our study is the recognition of the importance of all error metrics, each carrying its own set of advantages and disadvantages. The study emphasizes the significance of error magnitude, capturing

overlapping information, and clustering metrics that target similar error aspects. This allows us to streamline the calculation of the Bergen metric by avoiding redundant contributions from metrics in the same cluster.

In a recent HESS paper, spatial patterns of GCM/RCMs are used to the ranking.

https://doi.org/10.5194/hess-23-4803-2019

We have cited this paper in our modified draft.

Applying cluster analysis doesn't add much to the novelty of the metric. Each component must add something to the Bergen Metric.

We appreciate the reviewer's input. The primary objective of our study is to delve into the characteristics of error metrics, examining how they address diverse aspects of the relationship between observed and reference data errors. The disparity in error magnitudes and model rankings, as demonstrated in the study, can potentially lead to confusion among users. With an abundance of metrics available in the literature, our study does not seek to introduce a new one but rather aims to streamline the existing metrics by data analysis. Clustering plays a pivotal role in this process, especially when dealing with metrics that, despite targeting similar error types, exhibit significant data-oriented variations. Take, for instance, various versions of root mean squared error – in the absence of outliers, they remain in the same cluster, while the presence of outliers assigns them to different clusters. The study operates under the assumption that all error metrics are crucial. By integrating them into a composite metric without overlap (achieved through clustering) and subsequently analyzing individual error metrics, we can gain enhanced insights into the relationship between observed and modeled data.

One of the components in Eq7 should be histogram match (overlap %) for better discrimination power of the metric.

We appreciate the valuable feedback provided by the reviewer. Acknowledging the significance of histogram matching as an essential metric for model evaluation, we would like to highlight that Equation 7 (Bergen metric) has a flexible framework. This flexibility allows the incorporation of new metrics into the evaluation process. Recognizing the abundance of metrics available in the literature, the inclusion of additional metrics in the clustering analysis is anticipated to enhance our understanding of model performance.

Also the metric should be insensitive to the unit-differences since observation and simulation may have different units. GCM to GCM comparison can be smooth and not unit issue; however, observed AET from MODIS is watt/m2 whereas hydrologic model flux simulations are in mm/day. Then ,Bergen metric cannot be applied to other hydroclimatologic problems.

We appreciate the reviewer's comment. Firstly, it's important to note that the Bergen metric is unitless, as the error metrics used in its computation are normalized across all models. This characteristic enables its application to any hydroclimatologic problem, provided that the necessary error metrics can be calculated.

The unit of Actual Evapotranspiration (AET) is mm/day, while the latent heat flux has a unit of watt/m², which is derived from various remotely sensed data. The linear relationship between latent heat flux and water flux (AET) involves a constant, which is the product of the

latent heat of vaporization of water and water density. The interchangeability of units is possible due to this linear relationship. However, if the units are not linearly related, it could lead to complex issues, as there would be no basis for comparison.

It's worth noting that not all metrics need to be independent of units. Many widely used error metrics in climate studies, such as Mean Squared Error and Root Mean Squared Error, are unit-dependent. Both unit-dependent and unit-independent error metrics have their advantages and disadvantages, as highlighted by Hyndman (2006). Therefore, in this study, we consider all error metrics to be important and suggest incorporating them based on the specific requirements and underlying data of individual studies.

Hyndman, R. J. (2006). Another look at forecast-accuracy metrics for intermittent demand. Foresight: The International Journal of Applied Forecasting, 4(4), 43-46.

EOF, SSIM, FSS, SPEM and SPAEF metrics should be covered in the literature review.

We thank the reviewer for the comment. These methods have been referenced in the modified draft.

'In addition to this, researchers have employed various characteristics of climatic parameters as measures to assess and compare climate models with observed datasets. Metrics encompassing the frequency of days with precipitation over 1 mm and over 15 mm, the 90% quantile of the frequency distribution, and the maximum number of consecutive dry days, along with parameters such as daily mean, daily maximum, daily minimum, yearly maximum, length of the frost-free period, growing degree days ($> 5°C$), cooling degree days ($> 22°C$), heating degree days ($< 15.5°C$), days with RR ($>$ 99th percentile of daily amounts for all days), ratio of spatial variability, pattern correlation, ratio of interannual variability, temporal correlation of interannual variability, number of summer days, number of frost days, consecutive dry days, and ratio of yearly amplitudes, have been utilized for the validation of Euro-CORDEX data (Kotlarski et al., 2014; Giot et al., 2016; Smiatek et al., 2016; Torma, 2019; Vautard et al., 2021). Other studies have employed the empirical orthogonal functions (Rasmus et al., 2023), structural similarity index metric (Wang & Bovik, 2002), fractions skill score (Roberts & Lean, 2008), spatial pattern efficiency metric (Dembélé et al., 2020), spatial efficiency metric (Demirel, 2018) and probability distribution function (Perkins et al., 2007; Boberg et al., 2009; Boberg et al., 2010; Masanganise et al., 2014) to evaluate climate models.'

---

## Author Comment (AC4)

**Bergen Metrics: composite error metrics for assessing performance of climate models using EURO-CORDEX simulations**

Alok K. Samantaray, Priscilla A. Mooney, Carla A. Vivacqua

We appreciate the comments from the reviewers as they have made some important points that have been addressed in the revised manuscript.

The paper proposes a new methodology to obtain the best model evaluation, based on error metrics. It uses 38 different metrics, and through a clustering algorithm, it reduces the relevant metrics to less than half and then computes a composite metric to summarise the result. However, the paper does not demonstrate that this methodology allows for a ranking of the different models to build an informed ensemble mean. Is the Bergen Metric reasonably different from model to model or does it provide very similar results?

We thank the reviewer for the comment. In the revised manuscript, we have added a figure showing the inter-model variability of Bergen metric at a random grid point. The figure also shows the ranking of different models based on Bergen metric. The following analysis and the figure have been added to the text.

'To illustrate intermodel variability, a random grid point (50.125, 1.875) is selected. The Bergen metric is calculated for both precipitation and temperature at this grid point, and models are ranked based on the Bergen metric (Fig. 18). The Bergen metric ranges from 2.29 to 11.39 for precipitation and 1.85 to 8.37 for temperature. Notably, with a Bergen metric value of 2.29, ETH-COSMO (Model 6) is identified as performing well for precipitation. Similarly, with a Bergen metric value of 2.29, GERICS-REMO2015 (Model 16) is recognized for its good performance in temperature. The proposed metric offers a valuable tool for assessing the performance of climate models.'

[Figure]

**Figure R1:** The Bergen metric for precipitation (a) and temperature (b) for all 89 climate models, along with the ranking of each model based on the Bergen metric for precipitation (c) and temperature (d), at a grid point (50.125, 1.875).

Although it uses 38 different metrics, they all assume that the best model would be the one that has the best temporal synchronicity with the observations. While this is mostly true for the EURO-CORDEX evaluation simulations since they were forced by ERA-Interim, this is not the case for the historical runs. The latter were forced by GCMs and are only supposed to represent the historical climate.

We acknowledge that the RCM simulations driven by ERA-Interim exhibit superior temporal synchronicity compared to those driven by GCMs. Any error metrics such as RMSE or Bias may oversimplify the evaluation process by reducing it to a single numerical value, potentially overlooking deficiencies in specific model components or processes. It is crucial to underscore that our proposed metric evaluates the magnitude differences between modeled and reference data, prioritizing this aspect over spatial patterns. The application of this metric should be approached with careful consideration. This discussion has been added in the modified draft.

The introduction is also missing other types of performance metrics, which should also be added. E.g. metrics that evaluate the performance of the pdf or cdf are missing. Expand paragraphs 118 -121.There are several model performance studies of the EURO-CORDEX domain, either for the entire domain or, specific regions. The different metrics employed should be referenced.

We thank the reviewer for the comment. The following paragraph has been added to in the modified draft.

'In addition to this, researchers have employed various characteristics of climatic parameters as measures to assess and compare climate models with observed datasets. Metrics encompassing the frequency of days with precipitation over 1 mm and over 15 mm, the 90% quantile of the frequency distribution, and the maximum number of consecutive dry days, along with parameters such as daily mean, daily maximum, daily minimum, yearly maximum, length of the frost-free period, growing degree days ($> 5°C$), cooling degree days ($> 22°C$), heating degree days ($< 15.5°C$), days with RR ($> 99$th percentile of daily amounts for all days), ratio of spatial variability, pattern correlation, ratio of interannual variability, temporal correlation of interannual variability, number of summer days, number of frost days, consecutive dry days, and ratio of yearly amplitudes, have been utilized for the validation of Euro-CORDEX data (Kotlarski et al., 2014; Giot et al., 2016; Smiatek et al., 2016; Torma, 2019; Vautard et al., 2021). Other studies have employed the empirical orthogonal functions (Rasmus et al., 2023), structural similarity index metric (Wang & Bovik, 2002), fractions skill score (Roberts & Lean, 2008), spatial pattern efficiency metric (Dembélé et al., 2020), spatial efficiency metric (Demirel, 2018) and probability distribution function (Perkins et al., 2007; Boberg et al., 2009; Boberg et al., 2010; Masanganise et al., 2014) to evaluate climate models.'

The code provided does not allow for the reproduction of the manuscript's results. The input data for the clustering code should be provided as well as the code to compute the final Bergen Metric.

R1-4: We appreciate the reviewer's feedback. The code has been enhanced for improved usability and has been uploaded to Zenodo (https://doi.org/10.5281/zenodo.10518064). Unfortunately, the data cannot be provided and hosted by another user due to copyright issues. However, the data is openly available on the Earth System Grid Federation server, and the models used in the study are listed in Table S2. Users can download any models from the server and calculate the Bergen metric using the provided code.

A diagram with the complete methodology would be useful to better understand the different sections of the work.

We thank the reviewer for the comment. We have prepared a flow chart and it has been added in the revised draft (Figure 4).

[Figure]

Figure R2: The flowchart for the calculation of Bergen metric

The supplementary text S1 and S2 although relevant to the paper, are not referenced in the main text. Please add the corresponding references.

We thank the reviewer for the comment. It has been corrected in the revised draft.

Lines 153 – 162 The Euclidean distance framework is not clearly explained. Please rephrase

The paragraph has been rephrased and added in the revised draft.

'The Euclidean distance framework has found increasing use in various fields, serving as an error function or metric in applications like model evaluation, parameter optimization, and classification problems. In essence, it calculates the straight-line distance between two points in the space, known as Euclidean distance. The Euclidean distance is essentially the second norm of a vector. Equation 1 represents the generalized form of the p-norm in an n-dimensional vector space, where $xi$ is the vector. When p is set to 2, it transforms into the Euclidean norm. In the context of time series data, if the vector $(x_i)$ represents the difference between observed data $(u_i)$ and model data $(v_i)$ i.e., $x_i = u_i - v_i$, then d is termed the Euclidean distance metric. Here, $i$ represents the time series data. It's important to note that root mean squared error and mean squared error are different variants of the Euclidean distance metric.

Furthermore, if the vector represents the difference between error metrics (correlation coefficient $[u_1]$, absolute error $[u_2]$ and root mean squared error $[u_3]$) and their ideal values $(v_{1:3})$, then d is referred to as the DISO index. In summary, the Euclidean distance framework offers a versatile approach applicable to various scenarios, providing valuable insights through different metrics and indices.'

Lines 186-189 Which version of E-Obs was used? There is already an E-Obs grid on 0.1° resolution, why wasn't this grid used? Add the justifying text to the manuscript.

We thank the reviewer for the comment. Upon careful examination, we acknowledge that different versions employ varying numbers of observed weather stations. It is noteworthy, however, that the number of observed stations remains consistent between the 0.25-degree and 0.11-degree products, with the latter representing the higher interpolated dataset.

The selection between a 0.25-degree and a 0.11-degree resolution grid is depend upon the specific requirements of the study. In the context of this research, it forms a component of a broader investigation where its outcomes will be compared with the results derived from a 0.44-degree resolution Euro-CORDEX dataset, subsequently interpolated to the 0.25-degree resolution. It is essential to emphasize that the framework presented in this study exhibits independence from any specific resolution. This study serves as an example, showcasing the framework's versatility in evaluating different climate models. The discussion has been added in the revised manuscript.

Lines 198-206 belong in the introduction.

The paragraph has been relocated to the introduction section.

Line 209: Please provide the equations for each metric and the relevant references.

We appreciate the reviewer's feedback. The metrics employed in this study have been extensively detailed, including equations and references, in Jackson et al. (2019). The sentence has been revised for greater clarity in the updated manuscript.

Table 1: the order of the last columns should be changed so that it matches the text.

After examination, we observed that the order of the last column aligns with the text. The ranking order in both the table and the text is [2, 3, 1].